# SmolLM2: When Smol Goes Big –
# Data-Centric Training of a Fully Open Small Language Model

**Loubna Ben Allal,**[*] **Anton Lozhkov,**[*] **Elie Bakouch,**[*] **Gabriel Martín Blázquez,**[*]
**Guilherme Penedo, Lewis Tunstall, Andrés Marafioti, Hynek Kydlíček,**
**Vaibhav Srivastav, Joshua Lochner, Clémentine Fourrier, Hugo Larcher,**
**Ben Burtenshaw, Haojun Zhao, Caleb Fahlgren, Mathieu Morlon,**
**Agustín Piqueres Lajarín, Xuan-Son Nguyen, Cyril Zakka, Colin Raffel,**
**Leandro von Werra, Thomas Wolf**
🤗 Hugging Face

⌂ Code: https://github.com/huggingface/smollm
🤗 Models and datasets: https://huggingface.co/HuggingFaceTB

## Abstract

Large language models, while groundbreaking, are computationally expensive and difficult to deploy in resource-constrained settings. To address this challenge, small language models have emerged, but their performance critically depends on the quality and composition of the pretraining datasets—yet many recent models, such as Qwen2.5-1.5B and Llama3.2-1B, remain opaque about their training data, limiting reproducibility and scientific understanding. In this paper, we document and publicly release SmolLM2, a fully transparent state-of-the-art "small" (1.7 billion parameter) language model (LM), along with its training datasets and code. To attain strong performance, we overtrain SmolLM2 on ~11 trillion tokens of data using a multi-stage training process that mixes web text with specialized math, code, and instruction-following data. We additionally curate and release new specialized datasets (FineMath, Stack-Edu, and SmolTalk) at stages where we found existing datasets to be problematically small or low-quality. To inform our design decisions, we perform both small-scale ablations and a manual refinement process that updates the dataset mixing rates at each stage based on the performance at the previous one. Ultimately, we demonstrate that SmolLM2 outperforms other recent small LMs including Qwen2.5-1.5B, Llama3.2-1B, and Falcon3-1.6B. By releasing our model, datasets, and code, we aim to facilitate future research on LM development as well as applications of small LMs.

## 1 Introduction

Large language models (LMs) have become a cornerstone of modern AI systems due to their ability to follow natural language instructions and flexibly perform a huge range of tasks (Touvron et al., 2023; Bai et al., 2023; Brown et al., 2020; Dubey et al., 2024; Groeneveld et al., 2024; Chowdhery et al., 2023; Young et al., 2024; Taylor et al., 2022). LLMs are, by their nature, *large*, with many parameters (more than ~10 billion, by current conventions). This enormity results in high computational costs, both during training and for inference, which can prevent LLMs from being used in resource-constrained settings. To address this issue, a flurry of recent work has focused on efficient *small* (~3 billion parameters or less) LMs (Gunter et al., 2024; Yang et al., 2024b; AI@Meta, 2024b; Team et al., 2024; Li et al., 2023b). These small LMs are computationally inexpensive and can be run on a wider range of devices (e.g. mobile phones) while performing well on many important tasks.

A key factor in the performance and behavior of LMs is the data used to train them. While important for an LM of any size, data curation has an especially outsized influence for smaller models, as their limited capacity must be carefully optimized for learning core

---

[*]Equal contribution

knowledge and fundamental capabilities rather than memorizing incidental facts (Abdin et al., 2024a; Rolnick et al., 2017). Yet this aspect is often overlooked in recent small model releases (Yang et al., 2024b; AI@Meta, 2024a), which typically omit details about their exact training datasets and mixtures. This lack of transparency hinders both reproducibility and a deeper understanding of what drives small models performance.

This training data is primarily composed of text crawled from the web (Radford et al., 2019; Raffel et al., 2020), and state-of-the-art pipelines include sophisticated filtering and processing stages that aim to improve data quality (Li et al., 2024b; Penedo et al., 2024b;a; Soldaini et al., 2024). Recently, it has become common to include "specialized" data from certain domains such as software code (Kocetkov et al., 2022; Lozhkov et al., 2024) and mathematics (Paster et al., 2023; Han et al., 2024), which can improve performance not only on those domains but also more generally on challenging tasks that require reasoning (Muennighoff et al., 2023; Aryabumi et al., 2024). However, this full data is rarely released.

Motivated by the above, we aim to address the lack of transparency around pretraining data in small LMs by releasing a strong small model and its full training dataset. We also provide our training pipeline and methodology, by outlining all the steps that informed our decisions. First, we performed a careful evaluation of existing web, code, math, and instruction-following datasets (Section 3) to guide training data design choices. After finding that existing datasets were too small and/or low-quality, we created new state-of-the-art datasets: FineMath, Stack-Edu, and SmolTalk (for mathematics, code, and instruction-following respectively). We then trained a small LM (SmolLM2) via a multi-stage manual rebalancing of different sources to maximize performance (Section 4),[1] as well as its instruction-tuned variant (Section 5). Ultimately, we showed that both the base and instruction-tuned variants of SmolLM2 are state-of-the-art among similarly sized models (Section 4 and Section 5.3).

## 2 Related work

Modern LM training typically begins with "pretraining" on a large amount (e.g. trillions of tokens) of unstructured text. Pretraining helps the model fit language structure (Clark, 2019) and store factual knowledge (Petroni et al., 2019; Roberts et al., 2020) and therefore has proven to be a vital part of LM training, making pretraining dataset composition a key consideration. The data-hungry nature of pretraining has led to the use of large-scale web scrapes (com; ope; ant) which in their raw form can lead to poorly performing LMs (Penedo et al., 2024b). Consequently, the primary means of curation for modern LM pretraining datasets involves designing sophisticated pipelines for automatically filtering and reformatting web texts (Penedo et al., 2024a;b; Soldaini et al., 2024; Li et al., 2024b) that aim to keep enough data to avoid detrimental repetition (Muennighoff et al., 2023) while discarding any data that is not "high-quality". Apart from web text, including "specialized" data from certain domains – code (Kocetkov et al., 2022; Li et al., 2023a) and math (Paster et al., 2023; Han et al., 2024; Wang et al.) in particular – can improve model performance on tasks that involve reasoning and world knowledge (Muennighoff et al., 2023; Aryabumi et al., 2024; Lewkowycz et al., 2022; Shao et al., 2024). The contribution of small specialized datasets can be dwarfed by much larger web-based pretraining data sources, which has led to the design of multi-stage pretraining where specialized or high-quality datasets are incorporated later in training (Abdin et al., 2024b; Ai2, 2024; Blakeney et al., 2024).

Alternative approaches to training small language models from scratch, such as model distillation (Hinton et al., 2015), have been explored. However, these methods assume access to a larger, high-performing teacher model and its original training data—which is often unavailable for today's state-of-the-art LLMs. Moreover, distillation still requires training on trillions of tokens to achieve competitive performance (Team et al., 2024; AI@Meta, 2024b).

After pretraining, language models typically undergo two additional training stages before deployment: instruction tuning and preference learning. In instruction tuning, the model

---

[1]Such on-the-fly rebalancing is a promising for large-scale training runs which can be sufficiently costly (around 1e23 FLOPs, or $250,000 USD worth of GPU compute for SmolLM2) to preclude running multiple full-scale training runs.

undergoes supervised training on instruction/response pairs that reflect how it should answer queries (Wei et al., 2021; Mishra et al., 2021; Sanh et al., 2021; Wang et al., 2022). This process provides a valuable way of tailoring LMs to provide helpful responses rather than simply attempting to continue the input (as taught during pretraining). During preference learning, language models are further "aligned" towards their intended use by being trained to distinguish between helpful and unhelpful responses (Ouyang et al., 2022; Bai et al., 2022). This final stage typically involves a form of reinforcement learning (Bai et al., 2022; Lee et al.; Rafailov et al., 2024) on data labeled with human or synthetically generated preferences.

## 3    Pretraining datasets

Pretraining data curation is especially important for small LMs due to their tendency to be more sensitive to noise in the training data (Rolnick et al., 2017; Abdin et al., 2024a). In addition, designing a pretraining strategy involves not only selecting and curating data, but also determining how much to "mix" (i.e. sample) from different sources, which can be particularly important when including e.g. specialized math and code datasets. To address these challenges, we carefully evaluated existing datasets and created new ones where necessary. As a result, we release FineMath (mathematics) and Stack-Edu (code), two state-of-the-art datasets that outperform all existing open alternatives in their respective domains, and are designed to support the next generation of language models.

### 3.1    Ablation setup

To compare English web datasets and find the best mixture for training our models, we followed an empirical approach similar to Penedo et al. (2024a). Specifically, we trained models on each dataset under identical conditions: model configuration, training hyperparameters, and token count. We trained 1.7B parameter Transformers (Vaswani et al., 2017) based on the Llama architecture Touvron et al. (2023), with a sequence length of 2048, a global batch size of approximately 2 million tokens, the GPT-2 tokenizer Radford et al. (2019), and a cosine learning rate schedule (Loshchilov & Hutter, 2016) with a learning rate of $3.0 \times 10^{-4}$. Each dataset ablation model is trained on 350B tokens randomly sampled from the full dataset. For evaluation, we also followed Penedo et al. (2024a), and used `lighteval` to evaluate on a variety of knowledge, reasoning, and text understanding benchmarks: MMLU Hendrycks et al. (2021), HellaSwag Zellers et al. (2019), OpenBook QA Mihaylov et al. (2018), PIQA Bisk et al. (2019), WinoGrande Sakaguchi et al. (2019), ARC Clark et al. (2018), and CommonSenseQA Talmor et al. (2019).

Math and code capabilities typically emerge only after extensive training, so similarly to Blakeney et al. (2024); Dubey et al. (2024); Ai2 (2024), when evaluating math and code datasets we started from a mid-training checkpoint of SmolLM2 at 3T tokens (detailed in Section 4), which was trained primarily on web data. We then used an *annealing*: linearly decaying the learning rate to 0 while training on a mixture including the target dataset. For math, we annealed on a mixture of 60B tokens of the dataset under evaluation and 40B from the pre-checkpoint mixture. For code ablations, we performed annealing on 200B tokens, uniformly distributed across 15 of the most commonly used programming languages (~14B tokens each). We evaluated the math ablation models on GSM8K Cobbe et al. (2021), MATH Hendrycks et al. (2021) and MMLU-STEM to assess their math capabilities using `lighteval`, and we used HumanEval Chen et al. (2021) and MultiPL-E Cassano et al. (2022) to evaluate the code ablation models using the `BigCode-Evaluation-Harness`.

### 3.2    English web data

Web text from Common Crawl remains a popular source of pretraining data, and recent classifier-based filtering has significantly advanced pretraining data quality (Dubey et al., 2024; Abdin et al., 2024b;a; Kong et al., 2024). Two notable open datasets using classifer-based filtering are FineWeb-Edu (Penedo et al., 2024a) and DCLM (Li et al., 2024b). FineWeb-Edu contains 1.3T "educational" tokens identified by a classifier trained on annotations generated by Llama3-70B-Instruct (Dubey et al., 2024). DCLM comprises 3.8T tokens filtered with a fastText classifier (Joulin et al., 2016a;b) trained on OpenHermes 2.5 (Teknium, 2023a) instruction data and top-rated posts from the r/ExplainLikeImFive (ELI5) subreddit. Train-

ing ablation models on 350B tokens each from FineWeb-Edu and DCLM yields the results in Table 5 (Appendix A.3). FineWeb-Edu achieves higher scores on the educational benchmarks MMLU, ARC, and OpenBookQA, while DCLM performs better on HellaSwag and CommonsenseQA. These results align with the datasets' content: FineWeb-Edu prioritizes educational material, while DCLM captures more diverse, conversational styles.

Given the complementary strengths of FineWeb-Edu and DCLM, we explored mixing them and found that a 60% FineWeb-Edu and 40% DCLM mix works well, as shown in Table 5 (Appendix A.3): It nearly matches FineWeb-Edu's performance on MMLU and ARC while also aligning with DCLM's results on HellaSwag and approaching its performance on CommonSenseQA. Combining these datasets yields 5.1T tokens of (English) text.

## 3.3 Math data

Specialized math pretraining data is crucial for developing robust mathematical understanding. Recent research has shown that carefully curated mathematical content from Common Crawl, combined with targeted filtering techniques, can significantly enhance language models' mathematical reasoning capabilities (Dubey et al., 2024; Yang et al., 2024c; Shao et al., 2024; Han et al., 2024).

**Comparison of Existing Datasets**  We compare two leading publicly available math datasets: OpenWebMath (OWM) (Paster et al., 2023) and InfiMM-WebMath (Han et al., 2024). OWM consists of 12B tokens, built by filtering math-specific content from Common Crawl and using a specialized text extraction pipeline to preserve mathematical formatting and equations. InfiMM-WebMath contains 40B text tokens, and its authors show that it matches the performance of the private dataset of DeepSeekMath (Shao et al., 2024).

We ran annealing ablations (as described in Section 3.1) on OWM and InfiMM-WebMath: InfiMM-WebMath achieves a peak accuracy of 14% on GSM8K compared to OWM's 10%, while OWM slightly outperforms InfiMM-WebMath on MATH. The full evaluation curves are available in Appendix A.4.1. Despite training on 60B math tokens (i.e., 5 epochs for OWM and 1.5 epochs for InfiMM-WebMath), performance still lagged behind proprietary state-of-the-art small models (Yang et al., 2024b). Further analysis revealed two limitations: insufficient dataset sizes, and insufficient focus on step-by-step mathematical reasoning, along with an overrepresentation of academic papers that focus on advanced concepts.

**New dataset: FineMath**  The aforementioned issues with OWM and InfiMM-WebMath motivated us to develop FineMath, a collection of up to 54B tokens of math data focusing on mathematical deduction and reasoning through classifier-based filtering.

We extracted text from Common Crawl WARC files using Resiliparse, focusing on all 5.8B unique URLs from the FineWeb dataset (a subset of Common Crawl's 75B unique URLs). We then employed the FineWeb-Edu filtering approach, using Llama-3.1-70B-Instruct (Dubey et al., 2024) with a prompt (Appendix A.4.4) that scores content on a 3-point scale, where 1 indicates some mathematical content and 3 indicates step-by-step problem solutions at an appropriate level. We trained a classifier on these silver labels and identified domains containing at least 10 pages with a score of 2 or higher. We expanded our coverage by including domains with at least 10 URLs from either OWM or InfiMM-WebMath. From the Common Crawl index, we retrieved a total of 7.7B URLs belonging to this list of domains: 5.7B identified by our classifier, 0.6B from OWM, and 1.3B from InfiWebMath. We then re-extracted all identified pages using the OWM pipeline, preserving LaTeX formatting and removing all-boilerplate pages, yielding 7.1B pages containing 6.5T tokens.

To retain only high-quality math content, we reapplied a classifier trained on Llama-3.1-70B-Instruct annotations using a 5-point scale prompt (Appendix A.4.5) targeting pages with reasoning and middle- to high-school-level content. We note that InfiMM-WebMath used a similar classifier filtering technique, but it did not target the same type of content. After classification, we deduplicated using single-band MinHash LSH (Broder, 1997) with 10 hashes and applied fastText language classification (Joulin et al., 2016a;b) to keep only English content. Ultimately, we developed multiple variants of FineMath: FineMath4+ (10B tokens, 6.7M documents) which retains only samples with scores of 4-5 and FineMath3+ (34B tokens, 21.4M documents) which includes scores 3-5. We also applied the same

classifier to InfiMM-WebMath, creating Infi-WebMath4+ (8.5B tokens, 6.3M documents) and Infi-WebMath3+ (20.5B tokens, 13.9M documents). Similarly to Yang et al. (2024c), we decontaminate each dataset against GSM8K, MATH and MMLU using 13-gram matching and a minimum overlap ratio with the longest common subsequence of 0.6.

Figure 6 (Appendix A.4.2) shows FineMath ablations. All FineMath subsets outperform OWM and InfiMM-WebMath on GSM8K, MATH, and MMLU-STEM. FineMath4+ yields 2× higher GSM8K and 6× higher MATH scores compared to InfiMM-WebMath, demonstrating the importance of high-quality math with reasoning. Infi-WebMath4+ also outperforms InfiMM-WebMath but plateaus after 80B tokens (roughly 10 epochs), likely due to data repetition, a trend not in FineMath4+. To further validate FineMath, we additionally ran evaluation by doing continual pretraining of Llama3.2 3B base AI@Meta (2024b) on the above datasets. FineMath4+ yields the strongest improvements, boosting GSM8K by +20.5 points and MATH by +15.6 points compared to the original model, see Appendix A.4.3.

## 3.4 Code data

Code generation and understanding are becoming essential capabilities for modern LLMs, enabling diverse use cases such as code completion, debugging, and software design. While specialized code models (Lozhkov et al., 2024; Bai et al., 2023; Roziere et al., 2023) are optimized specifically for these tasks, general-purpose LLMs are increasingly deployed as coding assistants. Moreover, recent research has shown that including code data in pretraining enhances not only code-related capabilities but also improves natural language reasoning and world knowledge (Aryabumi et al., 2024). The Stack datasets are state-of-the-art open code datasets (Li et al., 2023a; Kocetkov et al., 2022), including Stack v1, ~3TB of source code from public GitHub repositories; StarCoderData (Li et al., 2023a; Kocetkov et al., 2022; Lozhkov et al., 2024), a filtered subset of 250 billion tokens across 80 programming languages; Stack v2, with ~32TB of data sourced from the Software Heritage code archive; and StarCoder2Data, the training corpus for StarCoder2 models (Lozhkov et al., 2024) with 900 billion tokens spanning more than 600 programming languages.

**Stack-Edu** Recent work has shown that the FineWeb-Edu classifier-based filtering strategy can be effective for code data (Wei et al., 2024b; Allal et al., 2024). We therefore constructed Stack-Edu, a filtered variant of StarCoder2Data focusing on educational and well-documented code. Specifically, we selected the 15 largest programming languages from StarCoder2Data to match the capacity constraints of smaller models (Lozhkov et al., 2024) and ensure benchmark coverage for the ablations. This subset had ~450 billion tokens. We then trained 15 language-specific classifiers using the StarEncoder model (Li et al., 2023a) on synthetic annotations generated by Llama3-70B-Instruct (Dubey et al., 2024) (prompt in Appendix A.5.1), which rated the educational quality on a scale from 0 to 5. Each classifier was trained on 500,000 samples and achieved an F1 score above 0.7 for most languages when applying a threshold of 3 for binary classification.

To evaluate Stack-Edu, we performed annealing ablations as described in Section 3.1. Filtering with a threshold of 3 improved performance across most languages while maintaining sufficient data, although Java performed better with threshold 2. Since Markdown is not included in the MultiPL-E benchmark, we could not determine a threshold for the dataset quantitatively; instead, we used threshold 3 based on qualitative analysis. Additionally, the base StarCoder2Data we started from was already decontaminated against MultiPL-E (Lozhkov et al., 2024). The resulting Stack-Edu dataset contains ~125B tokens across its 15 languages (see Appendix A.5.2). Table 1 shows the statistics of the top 4 programming languages in terms of size, and the positive impact of our educational filtering on MultiPL-E.

## 4 Pretraining

Recent trends in language models pretraining show a clear shift towards significantly longer training durations, especially for smaller models (Yang et al., 2024a;b; AI@Meta, 2024b). While this strategy deviates from the Chinchilla-optimal guidelines (Hoffmann et al., 2022), the resulting performance gains and reduced inference costs make extended training a worthwhile trade-off (de Vries, 2023). For example, Qwen2-1.5B was trained on 7 trillion tokens, Qwen2.5-1.5B on 18 trillion tokens, and Llama3.2-1B, derived from a pruned 8B

Table 1: Stack-Edu tokens counts for the top 4 largest languages with MultiPL-E ablation scores before (Orig.) and after (Fil.) filtering SC2Data (StarCoder2Data).

| Lang. | SC2Data (B) | Stack-Edu (B) | MultiPL-E (Orig.→Fil.) |
|---|---|---|---|
| Python | 50.6 | 21.8 | 20.7 → 25.6 |
| C++ | 69.7 | 16.0 | 16.7 → 24.8 |
| JS | 45.3 | 11.1 | 18.2 → 22.4 |
| Java | 45.6 | 42.1 | 17.6 → 22.7 |

Table 2: Performance **after each training stage**. S1–S3 are in the stable phase. Know. = knowledge/reasoning, Gen. = generative tasks. Benchmark results in Appendix A.6.1.

| Tokens | Stage 1 6T | Stage 2 8T | Stage 3 19T | Stage 4 11T |
|---|---|---|---|---|
| Know. | 55.5 | 56.8 | 57.5 | **60.2** |
| Math | 3.2 | 3.7 | 7.3 | **22.1** |
| Code | 8.9 | 10.6 | 16.7 | **23.2** |
| Gen. | 31.5 | 31.3 | 34.7 | **36.1** |

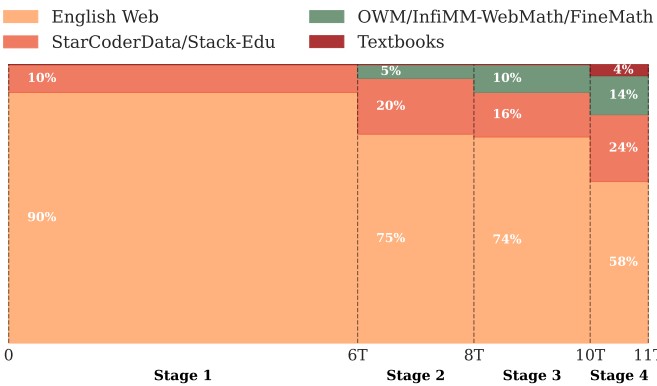

Figure 1: Dataset mixtures across training stages. Detailed descriptions are provided in Section 4. The x-axis represents the number of training tokens.

model, was trained using distillation on 9 trillion tokens (Yang et al., 2024a;b; AI@Meta, 2024b).

When building SmolLM2, we trained on 11 trillion tokens (approximately two epochs on our collected datasets), employing a multi-stage training approach instead of a fixed dataset mixture throughout pretraining. This design was guided by four key principles: (1) **Performance-driven interventions**, where we monitor evaluation metrics on key benchmarks and adapt dataset mixtures to address specific capability bottlenecks; (2) **Upsampling high-quality math and code during the annealing phase**, reserving datasets like FineMath and parts of Stack-Edu for the final stages to maximize their impact (Blakeney et al., 2024; Ai2, 2024); (3) **Strategic introduction of medium-sized datasets**, such as OWM, InfiMM-WebMath, and Stack-Edu, mid-training to avoid dilution by larger datasets early on; and (4) **Avoiding excessive data repetition**, in line with Muennighoff et al. (2023) we aimed to stay close to the recommended 4–5 epoch threshold for most datasets. While it might be fruitful to perform multiple from-scratch training runs to explore different data mixing schedules, the high cost of pretraining SmolLM2 (around $250,000 USD of GPU compute) motivated our "online" approach. In the following sections, we describe each stage of the training process, detailing the dataset mixtures, the rationale behind our choices, and the observations that guided our interventions. While some decisions were informed by established findings in the literature, others were driven by empirical insights gathered during training. The data mixtures of the four pretraining phases are available in Figure 1.

**Training setup** Our base model contains 1.7B parameters and follows the LLama2 (Touvron et al., 2023) architecture. It was trained on 256 H100s with the `nanotron` framework, AdamW optimizer, and a Warmup Stable Decay (WSD) (Hu et al., 2024; Zhai et al., 2022) scheduler with $5.0 \times 10^{-4}$ learning rate; see Appendix A.2 for the full training details.

**Stable phase: stage 1** In SmolLM2's first pretraining phase (0–6T tokens), we designed the mixture using insights from English web ablations and prior work. We adopted a 60% FineWeb-Edu and 40% DCLM ratio (discussed in Section 2.2) for web data, which provided

an optimal balance between educational content and diverse, real-world Q&A-style data. For code data, following Aryabumi et al. (2024), we incorporated StarCoderData, consisting of 250B tokens across 80 programming languages, and limited it to 10% of the total mixture to ensure approximately 4 epochs over 11T tokens with room for upsampling in later stages. We did not include math data in stage 1 due to our math datasets' relatively small size.

After 6T tokens of training, we evaluated SmolLM2 on key benchmarks, as shown in Table 2. Knowledge and reasoning performance aligned with expectations based on the web ablations but we observed generally poor coding and mathematics performance.

**Stable phase: stage 2**  For stage 2 (6T to 8T tokens), we added OWM at a 5% ratio and increased the proportion of code data to address coding and math gaps while preserving knowledge retention. The low OWM percentage reflects the dataset's small size (12B tokens) and our gradual math integration approach. The final mixture for stage 2 consisted of 75% English web data (keeping the 60/40 FineWeb-Edu to DCLM ratio from stage 1), 20% code data, and 5% math data, as shown in Figure 1.

After stage 2, code performance improved across most languages, validating the decision to upsample StarCoderData. OWM integration had no significant impact on math performance, highlighting the need for larger, higher-quality math datasets in later stages. Beyond code and math performance, we observed above-random (>25%) MMLU accuracy with a multiple-choice formulation (MCF) (Figure 7, Appendix A.6.2)), which is typically challenging for small models (Gu et al., 2024; Du et al., 2024). To further optimize MMLU performance, we revisited our English dataset mixture with additional annealing ablations and found that increasing DCLM relative to FineWeb-Edu slightly improves MMLU MCF at this stage.

**Stable phase: stage 3**  In the third and last stage of the stable phase (8T to 10T tokens, before annealing starts), we added the text-only English portion of InfiMM-WebMath with OWM, bringing the proportion of math to approximately 10%, as shown in Figure 1. For English web data, we revisited our ablations and adjusted the FineWeb-Edu to DCLM ratio to 40/60. For code, we replaced StarCoderData with Stack-Edu (Section 3.4) and used StarCoder2Data for languages with fewer than 4B tokens in Stack-Edu. We also added Jupyter Notebooks from StarCoder2 (Lozhkov et al., 2024), which provides rich, contextual examples of code interleaved with explanations, enhancing the model's reasoning around programming tasks. Adding these datasets brought improvements on most benchmarks.

**Decay phase: stage 4**  The final stage consisted of decaying the learning rate linearly to 0 for 10% of the total training duration (from 10T to 11T tokens) (Hägele et al., 2024). Following Blakeney et al. (2024), we introduced our highest quality mathematical datasets, InfiWebMath-3+, and FineMath 4+. We additionally allocated 0.08% of the mixture to OWM and 0.02% to AugGSM8K (Li et al., 2024a), an augmented version of the GSM8K benchmark's training set, which has become a common component of recent pretraining datasets (Achiam et al., 2023; Dubey et al., 2024; Ai2, 2024). Overall, mathematical content totaled 14% of the mixture. We expanded Stack-Edu to include additional programming languages not covered in stage 3, and set the dataset's contribution to 24% of the mixture. We maintained the natural distribution across programming languages, with a higher allocation for Python. The remaining mixture consisted of English web data at 58% (maintaining the higher DCLM to FineWeb-Edu ratio) and Cosmopedia v2 (Allal et al., 2024) at 4%, which provides 30B tokens of high-quality synthetic textbooks, blog posts, and stories.

While all the benchmarks show improvements after stage 4, we observe substantial gains in coding performance and, most notably, in math performance, validating our data mixture specifically targeting these domains. Additionally, performance metrics continued to improve up to the final stage as shown in Table 2 validating the decision to train SmolLM2 on 11T tokens for improved downstream performance.

**Context Length Extension**  We extended the context from 2k to 8k tokens during the final 75B tokens of training, following standard practice (Gao et al., 2024). Details in Appendix A.8.1. After this step, we obtain the final SmolLM2 base model.

**Base model evaluation**  We evaluate and compare the final base SmolLM2 model with existing state-of-the-art models of similar size, Qwen2.5-1.5B (Yang et al., 2024b), Llama3.2-

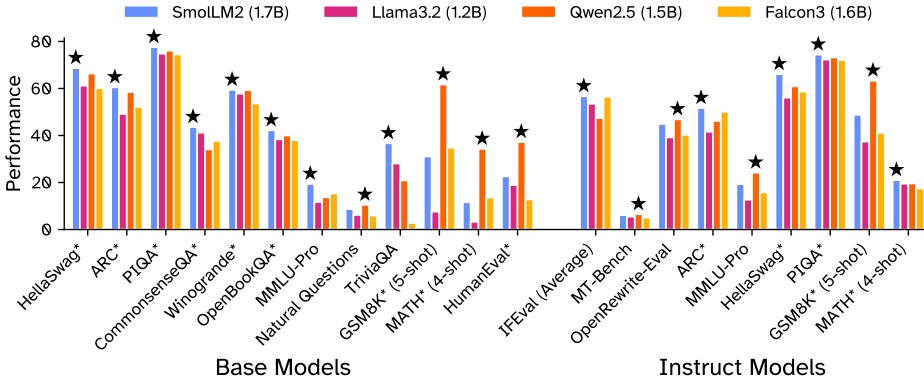

Figure 2: Performance comparison of SmolLM2 and other 1-2B base (left) and instruct (right) models across benchmarks. SmolLM2 outperforms other models on most benchmarks. The best model for each tasks is marked with a ★. Tasks used in ablation experiments are marked with *. Numerical results are provided in Table 9 and Table 12 (appendix).

1B (AI@Meta, 2024a) and Falcon3-1.6B (TII, 2024), on a wide range of benchmarks. Evaluations are conducted using `lighteval` and in a zero-shot setting unless otherwise specified. Evaluation results in Figure 2 show the strong performance of base SmolLM2, outperforming the Qwen2.5 base model on HellaSwag, and ARC. SmolLM2 also delivers strong performance on held-out benchmarks not monitored during training, such as MMLU-Pro (Wang et al., 2024), TriviaQA (Joshi et al., 2017), and Natural Questions (NQ, Kwiatkowski et al., 2019). Notably, the model outperforms Qwen2.5-1.5B by nearly 6 percentage points on MMLU-Pro, further validating its generalization capabilities.

On math and coding, SmolLM2 outperforms Llama3.2 on GSM8K, MATH, and HumanEval, and surpasses Falcon3 on HumanEval. While it trails Qwen2.5—trained on over 1T tokens of private math data, including Chinese and synthetic sources (Yang et al., 2024b)—our math corpus totals only 54B English tokens. Falcon3's training data is undisclosed. Despite this, FineMath and Stack-Edu represent the strongest publicly available datasets in their domains and are released to support future research on open high-quality pretraining data.

We see next to no degradation in performance after Context Length Extension, while the HELMET (Yen et al., 2024) and Needle in the Haystack (NIAH) (Kamradt, 2024) results show strong performance – see Appendix A.8.2. These results highlight the effectiveness of our curated datasets, data mixtures, and training stages.

## 5 Post-training

After training the base SmolLM2 model, we followed standard practice for maximizing performance and utility via post-training through instruction tuning and preference learning. We leveraged existing datasets in addition to a new dataset called SmolTalk.

### 5.1 SmolTalk

Although the SmolLM2 base model outperformed other state-of-the-art base models in the 1-2B parameter range, the base model's performance after fine-tuning on public datasets like MagPie-Pro (Xu et al., 2024) or OpenHermes2.5 (Teknium, 2023b) was lower than the post-trained versions of these other models. This observation motivated the development of SmolTalk, a new instruction-following dataset that carefully combines selected existing datasets with new synthetic datasets we developed, including the Magpie-Ultra conversational dataset as well as other task-specific datasets like Smol-Constraint, Smol-Rewrite, and Smol-Summarization. All datasets were generated using Distilabel (Bartolomé et al., 2024).

**Conversational data**  MagPie-Ultra is a multi-turn dataset created using the prompting method introduced in Xu et al. (2024), but leveraging Llama-3.1-405B-Instruct-FP8 (Dubey

et al., 2024) as the teacher model instead of Llama-3-70B-Instruct. We also incorporated system prompts to guide the generation, resulting in a balanced dataset of 1M three-turn conversations.

We compare MagPie-Ultra to existing public supervised fine-tuning (SFT) datasets in Table 13 (Appendix A.7). The evaluation suite included the instruction-following and conversation benchmarks IFEval (Zhou et al., 2023) and MT-Bench (Zheng et al., 2023); reasoning in ARC Challenge; knowledge in MMLU-Pro, GSM8K and MATH for math evaluations. Our dataset outperforms MagPie-Pro on most benchmarks, and largely surpasses OpenHermes2.5 and UltraChat (Ding et al., 2023) on IFEval and MT-Bench.

**Task-specific data**  We developed task-specific datasets to improve instruction-following with constraints (Smol-Constraint), summarization (Smol-Summarization), and rewriting (Smol-Rewrite). Smol-Constraint contains 36k instructions with detailed constraints. We generated over 500k instruction-response pairs using Qwen2.5-72B-Instruct (Yang et al., 2024b) and filtered them to remove conflicting constraints, incorrect responses, and IFEval contamination (10 n-gram overlap). For Smol-Summarization and Smol-Rewrite, we generated high-quality source texts—emails, tweets, LinkedIn posts, and notes—by prompting Qwen2.5-72B-Instruct with specific system prompts and personas from PersonaHub (Ge et al., 2024) and FinePersonas dataset (Argilla, 2024; Chan et al., 2024). Then we used the same model to produce  1M summaries and 600k rewritten versions. Adding the 3 Smol-datasets to MagPie-Ultra (MagPie-Ultra+) further improves IFEval performance as shown in Table 13 (Appendix A.7).

To boost math reasoning, we evaluated public math instruction datasets by fine-tuning on mixtures with 80% general instruction data (MagPie Ultra + Smol-Constraint, Smol-Rewrite, Smol-Summarization) and 20% math data from various sources. Results in Table 13 (Appendix A.7) show complementary dataset strengths: NuminaMath-CoT (Li et al., 2024c) performed well on MATH and MT-Bench, while MetaMathQA (Yu et al., 2023), which is also included in OpenHermes2.5, improved results on GSM8K. Based on these findings, we incorporated a combination of both datasets into SmolTalk.

**Other specialized data**  For code generation, we compared Self-OSS-Starcoder2-Instruct (Wei et al., 2024a), Code-Feedback (Zheng et al., 2024), and MagiCoder (Wei et al., 2023), selecting Self-OSS-Starcoder2-Instruct based on its HumanEval (Chen et al., 2021) performance. This dataset contains 50k high-quality Python instruction-response pairs. To support system prompts, we included 30k randomly selected samples from System-Chats2.0 (Computations, 2024), and for function calling, we added 80k samples from APIGen-Function-Calling (Liu et al., 2024). Additionally, to maintain strong performance on long-context tasks, we compared SEALONG (Li et al., 2024d) (a subset generated by Qwen2.4-15B-Instruct (Yang et al., 2024b)) and LongAlign (Bai et al., 2024) (English subset of 3.7k samples with 8k–16k tokens), selecting the latter as it provided benefits across most HELMET tasks. We also added 100k randomly selected OpenHermes2.5 samples due to its strong performance in knowledge (MMLU-Pro), Everyday-Conversations (Face, 2024), 2.2k casual multi-turn interactions, and Explore-Instruct (Wan et al., 2023) for rewriting. We found that incorporating these datasets with the specified number of samples effectively enhanced their target capabilities while preserving strong performance across other benchmarks.

## 5.2  Supervised fine-tuning and alignment

Table 10 (Appendix A.7) shows the final composition of SmolTalk. We performed supervised fine-tuning of our base SmolLM2 on SmolTalk for 2 epochs, using a global batch size of 128, sequence length of 8192, and a learning rate of $3.0 \times 10^{-4}$. The evaluation results after this SFT phase are available in Table 13 (Appendix A.7).

For preference learning, we used Direct Preference Optimization (DPO) (Rafailov et al., 2024). We experimented with various public synthetic feedback datasets (Ivison et al., 2024) including UltraFeedback (Cui et al., 2024), UltraInteract (Yuan et al., 2024), Capybara (Daniele & Suphavadeeprasit, 2023), and ORCA (Lv et al., 2023). UltraFeedback was the most effective, improving MT-Bench, MMLU-Pro, and MATH. We trained for 2 epochs

with a learning rate of $1.0 \times 10^{-6}$, beta 0.5, global batch size 128, and sequence length of 1024 tokens. After this stage, we obtain the instruct SmolLM2 model. As noted in Dubey et al. (2024), using short-context data for DPO did not impact the model's 8k context ability.

### 5.3 Instruct model evaluation

We evaluate the final instruct version of SmolLM2 and compare it with the instruct variants of Qwen2.5-1.5B, Llama3.2-1B and Falcon3-1.6B, with results shown in Figure 2. SmolLM2-Instruct shows strong instruction following capabilities, strongly outperforming Qwen2.5-1.5B-Instruct on IFEval; our model is competitive on MT-Bench and OpenRewrite-Eval (Shu et al., 2024) for text rewriting, and demonstrates strong mathematical capabilities as evidenced by the GSM8K and MATH scores. Additionally, SmolLM2 was submitted to the LMSYS Chatbot Arena leaderboard (Chiang et al., 2024) for human evaluation, where it achieved an Elo score of 1043 (95% CI: 1031–1058), comparable to Llama3.2-1B-Instruct's score of 1050 (95% CI: 1044–1056). These results highlight SmolLM2's ability to generalize across a variety of tasks, showcasing its potential as a capable chat assistant.

## 6 SmolLM2 135M and 360M

In addition to SmolLM2-1.7B, we also trained two smaller models: SmolLM2-360M (360M parameters, trained on 4T tokens) and SmolLM2-135M (135M parameters, trained on 2T tokens), which are similarly state-of-the-art in their size class. Given their smaller capacity and reduced training cost, we re-ran data ablations at the target training length to determine the most effective data mixture. We found that filtering DCLM with the FineWeb-Edu classifier, removing samples with score 0, and downsampling those with scores 1 and 2 worked best. Unlike SmolLM2-1.7B, where we leveraged a multi-stage training strategy, these smaller models benefited from a single-stage training approach with consistently high-quality data. We incorporated Stack-Edu from the start, alongside InfiMM-WebMath, FineMath, and Cosmopedia. These models share the same architecture as SmolLM2-1.7B but use Grouped Query Attention (GQA) and were trained using the WSD scheduler with 20% decay and a learning rate of $3.0 \times 10^{-3}$. For post-training, we applied SFT using a filtered version of SmolTalk[2], removing complex instruction-following tasks (e.g., function calling) and hard examples from MagPie-Ultra to better align with the models' capacity. Finally, we performed DPO training using UltraFeedback, optimizing the models for instruction-following while preserving coherence and helpfulness. More details about SmolLM2-360M and 135M can be found in their respective model cards[34].

## 7 Conclusion

SmolLM2 advances the state-of-the-art for open small LMs through a combination of careful dataset curation and multi-stage training. Our approach highlights the critical role of high-quality, specialized datasets in enabling smaller models to achieve strong performance across a variety of benchmarks. The development of FineMath, Stack-Edu, and SmolTalk (see Table 3) addressed limitations in existing public datasets, improving capabilities in reasoning, mathematics, and instruction-following tasks. While our math and code datasets are state-of-the-art among publicly available datasets, SmolLM2 still lags behind Qwen2.5 on these domains (despite shortening the gap after post-training). This underscores the importance of continued work on open, high-quality domain-specific datasets. To support future research, we release SmolLM2 alongside all the datasets and code used in its training, offering a comprehensive foundation for building performant small models.

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

# A  Appendix

## A.1  New open datasets

| Dataset | Samples | Tokens | Target Domain | Sources |
|---|---|---|---|---|
| **FineMath** | 35M | 54B | Math (pretraining) | Common Crawl |
| **Stack-Edu** | 167M | 125B | Code (pretraining) | GitHub (The Stack v2) |
| **SmolTalk** | 1.1M | 950M | Post-training | Synthetic |

Table 3: Summary of FineMath, Stack-Edu, and SmolTalk.

We developed and publicly released three specialized datasets to address gaps in existing publicly available training data. Table 3 summarizes these contributions:

- **FineMath**[5]: Large-scale mathematical reasoning dataset from Common Crawl for pretraining.
- **Stack-Edu**[6]: Curated coding dataset covering 15 programming languages from GitHub repositories.
- **SmolTalk**[7]: Synthetic conversational dataset for post-training, covering different domains including instruction following, chat, code and math.

## A.2  Training setup

Our base model contains 1.7B parameters and follows the LLama2 (Touvron et al., 2023) architecture, outlined in Table 4. We trained the model on 256 H100s using the nanotron framework and use AdamW optimizer with $(\beta, \beta_2) = (0.9, 0.95)$ with a Warmup Stable Decay (WSD) (Hu et al., 2024; Zhai et al., 2022) learning rate schedule to avoid setting a fixed training duration (see Figure 3). The schedule started with a 2,000-step warmup phase, maintained a peak learning rate of $5.0 \times 10^{-4}$ (stable phase), and could transition to a decay phase when needed, reducing the learning rate to zero over 10% of the total training steps (Hägele et al., 2024). We used the tokenizer from Allal et al. (2024), which has a vocabulary size of 49,152 tokens and was trained on a mixture of 70% of FineWeb-edu, 15% Cosmopedia-v2, 8% OpenWebMath, 5% StarCoderData and 2% StackOverflow.

Table 4 shows the architecture details of SmolLM2 1.7B.

Table 4: Overview of the architecture of SmolLM2. [†] This is before extending the context to 8k tokens.

| Parameter | Value |
|---|---|
| Layers | 24 |
| Model Dimension | 2,048 |
| FFN Dimension | 8,192 |
| Attention Heads | 32 |
| Sequence Length | 2,048 [†] |
| Token per batch | 2M |
| Tied embedding | Yes |
| Positional Embeddings | RoPE ($\theta = 10,000$) |
| Activation Function | SwiGLU |

Figure 3 shows the progression of the learning rate through the training using WSD scheduler.

---

[5] https://huggingface.co/datasets/HuggingFaceTB/finemath
[6] https://huggingface.co/datasets/HuggingFaceTB/stack-edu
[7] https://huggingface.co/datasets/HuggingFaceTB/smoltalk

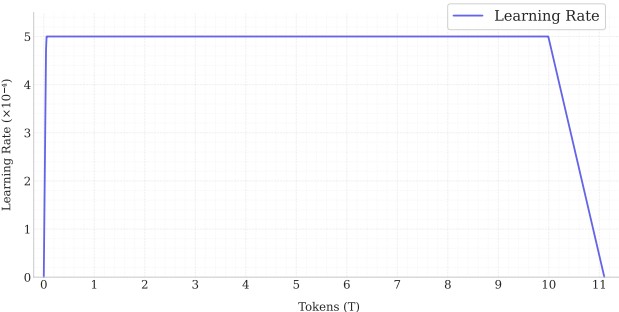

Figure 3: Learning rate during SmolLM2 training. We used WSD scheduler with 2000 steps warmup, learning rate $5.0 \times 10^{-4}$ and 10% decay.

Table 5: Evaluation of models trained on FineWeb-Edu and DCLM for 350B tokens. 40/60 and 60/40 denote the FW-Edu/DCLM ratio.

| Task | FW-Edu | DCLM | 40/60 | 60/40 |
|------|--------|------|-------|-------|
| MMLU | **37.5** | 35.5 | 36.5 | 37.0 |
| ARC | **57.5** | 53.5 | 53.2 | 56.0 |
| OpenBookQA | **41.9** | 40.8 | 39.0 | **41.9** |
| HellaSwag | 60.1 | **62.3** | 61.4 | 62.2 |
| CommonsenseQA | 36.2 | **40.1** | 39.9 | 38.5 |
| PIQA | 76.2 | **76.9** | 75.7 | 76.4 |

## A.3 English web ablations

Table 5 shows the evaluation results of ablation models trained on 350B tokens from DCLM, FineWeb-Edu and their mix. Figure 4 shows the evaluation of their intermediate checkpoints.

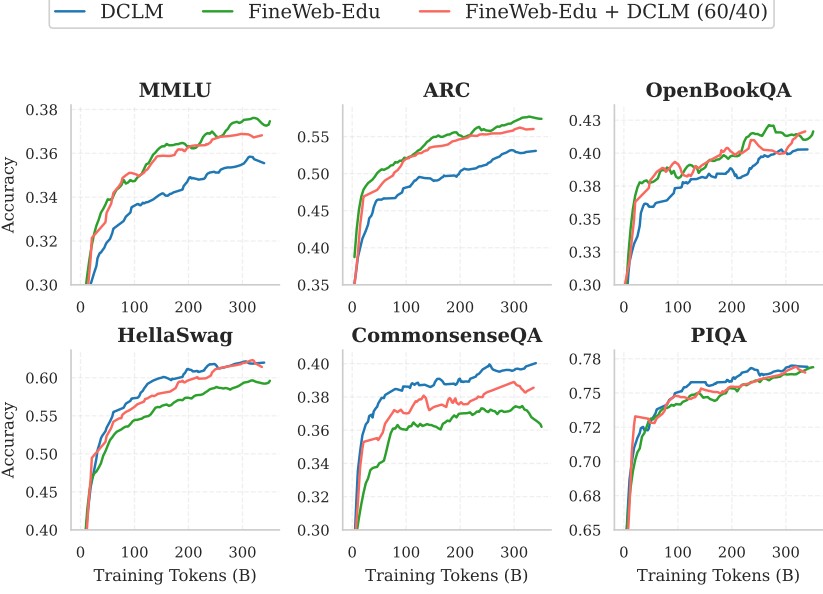

Figure 4: Evaluation of models trained on FineWeb-Edu and DCLM for 350B tokens. FineWeb-Edu excels at knowledge and reasoning tasks, while DCLM demonstrates stronger performance on commonsense reasoning benchmarks. A 60/40 mixture of FineWeb-Edu and DCLM achieves balanced performance across all tasks.

### A.4 FineMath

#### A.4.1 Public datasets comparison

Figure 5 Shows the performance of ablation models trained on OWM and InfiMM-WebMath on GSM8k and MATH.

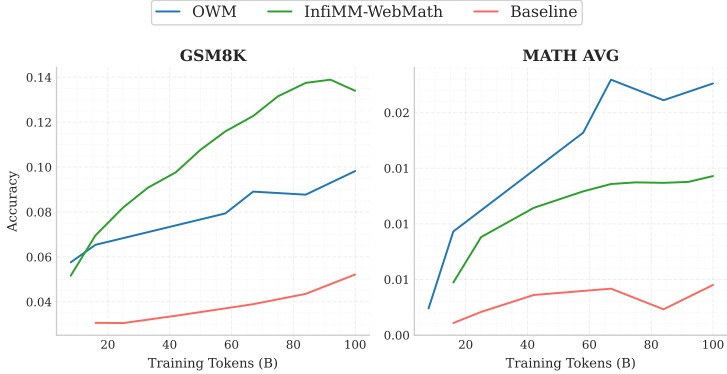

Figure 5: Results of annealing ablations comparing OWM and the text component of InfiMM-WebMath. InfiMM-WebMath consistently outperforms OWM on GSM8K, while OWM has a slight advantage on MATH. Despite training on 60B math tokens (equivalent to 5 epochs for OWM and 1.5 epochs for InfiMM-WebMath), performance remains far below state-of-the-art LLMs, highlighting the need for a new math dataset.

#### A.4.2 FineMath annealing ablations

Figure 6 shows the results of the SmolLM2 annealing ablations.

#### A.4.3 FineMath continual pretraining ablations

Table 6 shows the results of continual pretraining of Llama3.2 3B base (AI@Meta, 2024b) on 60B tokens from FineMath and other math datasets. These results further validate the effectiveness of FineMath.

#### A.4.4 Annotation Prompt (3-scale)

We used the following prompt template to generate the silver 3-scale annotations for FineMath using the Llama3 model:

> Evaluate the following text extract for its potential usefulness for studying mathematics up to high school and early undergraduate levels. Use the following 3-point scoring system described below. Points are accumulated based on the satisfaction of each criterion:

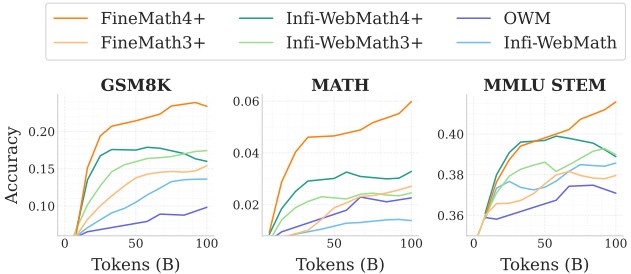

Figure 6: Performance of models trained on different subsets of FineMath and other math datasets.

Table 6: Continual pretraining of Llama3.2 3B on different math datasets (60B tokens each). FineMath variants, particularly FineMath4+, yield the strongest improvements on GSM8K and MATH, validating the effectiveness of our curation pipeline. The baseline corresponds to the original Llama3.2 3B base model without continual training.

| Dataset | GSM8K | MATH |
|---|---|---|
| Baseline | 26.7 | 8.0 |
| OWM | 34.0 | 14.8 |
| InfiMM-WebMath | 35.2 | 14.8 |
| Infi-WebMath3+ | 42.0 | 18.9 |
| Infi-WebMath4+ | 40.8 | 19.5 |
| FineMath3+ | 46.1 | 21.4 |
| FineMath4+ | **47.2** | **23.6** |

- Add 1 point if the extract contains some mathematical content, even if it's not very useful for studying or is an academic paper that is too advanced.

- Add another point if the extract demonstrates logical reasoning in a mathematical context, even if it lacks step-by-step explanations or is too advanced.

- Award a third point if the extract is at an appropriate level (up to high school and early undergraduate levels) and contains clear mathematical deductions and step-by-step solutions to mathematical problems.

Question-answer formats (e.g., from educational websites or forums) are acceptable if they meet the criteria. Ignore any formatting errors or missing equations and make assumptions based on the overall content.

The text extract:

<EXTRACT>

After examining the extract:

- Briefly justify your total score, up to 100 words.

- Conclude with the score using the format: "Final score: <total points>".

### A.4.5 Annotation Prompt (5-scale)

We used the following prompt template to generate the 5-scale annotations for FineMath using the Llama3 model during the second filtering stage:

Evaluate the following text extract for its potential usefulness for studying mathematics up to high school and early undergraduate levels. Use the following 5-point scoring system described below. Points are accumulated based on the satisfaction of each criterion:

- Add 1 point if the extract contains some mathematical content, even if it's not very useful for studying, or if it contains non-academic content such as advertisements and generated pages for converting weight and currencies.

- Add another point if the extract touches on mathematical topics, even if it's poorly written if it's too complex such as an academic paper that is too advanced.

- Award a third point if the extract demonstrates problem solving or logical reasoning in a mathematical context, even if it lacks step-by-step explanations.

- Grant a fourth point if the extract is at an appropriate level (up to high school and early undergraduate levels) and contains clear mathematical deductions and step-by-step solutions to mathematical problems. It should be similar to a chapter from a textbook or a tutorial.

- Give a fifth point if the extract is outstanding in its educational value for teaching and studying mathematics in middle school and high school. It should include very detailed and easy to follow explanations.

Question-answer formats (e.g., from educational websites or forums) are acceptable if they meet the criteria.

The text extract:

<EXTRACT>

After examining the extract:
- Briefly justify your total score, up to 100 words.
- Conclude with the score using the format: Final score: `<total points>`.

### A.5 Stack-Edu

#### A.5.1 Annotation Prompt

We used the following prompt template to generate the 5-scale annotations for Stack-Edu (Python in this case) using the Llama3 model:

> Below is an extract from a Python program. Evaluate whether it has a high educational value and could help teach coding. Use the additive 5-point scoring system described below. Points are accumulated based on the satisfaction of each criterion:
> - Add 1 point if the program contains valid Python code, even if it's not educational, like boilerplate code, configs, and niche concepts.
> - Add another point if the program addresses practical concepts, even if it lacks comments.
> - Award a third point if the program is suitable for educational use and introduces key concepts in programming, even if the topic is advanced (e.g., deep learning). The code should be well-structured and contain some comments.
> - Give a fourth point if the program is self-contained and highly relevant to teaching programming. It should be similar to a school exercise, a tutorial, or a Python course section.
> - Grant a fifth point if the program is outstanding in its educational value and is perfectly suited for teaching programming. It should be well-written, easy to understand, and contain step-by-step explanations and comments.
> The extract: `<EXTRACT>`
> After examining the extract:
> - Briefly justify your total score, up to 100 words.
> - Conclude with the score using the format: Educational score: `<total points>`

We use similar prompts for the other 14 programming languages in Stack-Edu, adjusting the examples in the third criterion to reflect language-specific topics. For instance, in the JavaScript prompt, we replace "deep learning" with "asynchronous programming".

#### A.5.2 Stack-Edu language statistics

Table 7 shows the size of each programming language in Stack-Edu before and after the educational filtering. Initially, we also included HTML, but the classifier performed poorly, so we retained StarCoder2Data.

Table 7: Stack-Edu dataset statistics across programming languages. The table shows the original dataset size (from StarCoder2Data) and filtered Stack-Edu size for each programming language.

| Language | StarCoder2Data (B tokens) | Stack-Edu (B tokens) |
|---|---|---|
| Python | 50.6 | 21.8 |
| Cpp | 69.7 | 16.0 |
| Markdown | 80.4 | 14.0 |
| C | 38.4 | 11.1 |
| JavaScript | 45.3 | 11.1 |
| Java | 45.6 | 42.1 |
| SQL | 13.7 | 9.62 |
| PHP | 44.9 | 9.07 |
| C-Sharp | 33.4 | 8.87 |
| TypeScript | 12.2 | 3.03 |
| Shell | 4.17 | 3.13 |
| Swift | 3.71 | 1.83 |
| Go | 3.67 | 1.80 |
| Rust | 3.39 | 1.75 |
| Ruby | 5.76 | 1.61 |

### A.6 Detailed pretraining results

#### A.6.1 Evaluation after each training stage

Table 8 shows the evaluation results of SmolLM2 at the end of each training stage. In addition to the benchmarks used during the ablations, we added four generative tasks: CoQA (Reddy et al., 2019), DROP (Dua et al., 2019), Jeopardy (MosaicML, 2024) and SQuAD v2 (Rajpurkar et al., 2018)

Table 8: Per-benchmark model performance across training stages. Stages 1-3 are during stable phase (no learning rate decay).

|  | Stage 1 | Stage 2 | Stage 3 | Stage 4 |
| Tokens | 0-6T | 6-8T | 8-10T | 10-11T |
| --- | --- | --- | --- | --- |
| MMLU (MCF) | 29.62 | 37.96 | 42.54 | **48.87** |
| HellaSwag | 66.17 | 65.29 | 66.29 | **69.26** |
| ARC | 59.95 | 60.08 | 58.66 | **60.99** |
| OpenBookQA | 42.00 | 42.40 | 41.40 | **43.60** |
| WinoGrande | 58.88 | 58.33 | 58.64 | **61.09** |
| PIQA | 76.39 | 76.50 | 77.26 | **77.64** |
| GSM8K | 4.32 | 4.62 | 10.01 | **32.60** |
| MATH | 2.1 | 2.78 | 4.52 | **11.54** |
| HumanEval | 10.97 | 9.15 | 17.68 | **22.60** |
| Multiple-E Java | 5.70 | 10.12 | 14.56 | **23.42** |
| Multiple-E JS | 9.94 | 12.42 | 18.01 | **23.60** |
| CoQA | 33.43 | 33.98 | 38.82 | **40.45** |
| DROP | 13.69 | 11.36 | 17.19 | **19.22** |
| Jeopardy | 23.1 | 22.4 | **25.54** | 23.35 |
| SQuAD v2 | 55.97 | 57.45 | 57.26 | **61.48** |

#### A.6.2 MMLU progression

Figure 7 shows the progression of MMLU scores throughout the stable phase.

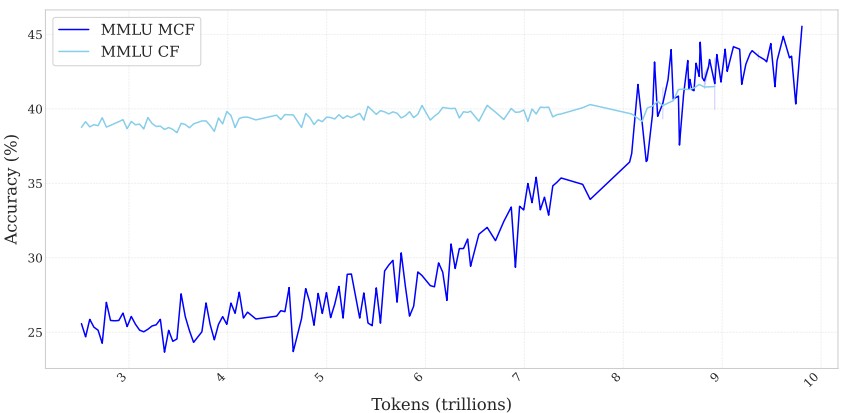

Figure 7: Progression of MMLU MCF and MLU CF during the training. We observe above-random (higher than 25%) accuracy on MMLU MCF after 6T tokens of training, while MMLU CF appears to plateau.

#### A.6.3 Base model comparison

Table 9 compares the performance of SmolLM2 after pre-training to other small base models.

Table 9: Performance comparison of SmolLM2 and other 1-2B **base** models across benchmarks. SmolLM2 outperforms other models on most benchmarks.

| Model family | SmolLM2 | Llama3.2 | Falcon3 | Qwen2.5 |
|---|---|---|---|---|
| Parameters | 1.7B | 1.2B | 1.6B | 1.5B |
| HellaSwag | **68.7** | 61.2 | 60.1 | 66.4 |
| ARC | **60.5** | 49.2 | 52.2 | 58.5 |
| PIQA | **77.6** | 74.8 | 74.5 | 76.1 |
| CommonsenseQA | **43.6** | 41.2 | 37.7 | 34.1 |
| Winogrande | **59.4** | 57.8 | 53.6 | 59.3 |
| OpenBookQA | **42.2** | 38.4 | 38.0 | 40.0 |
| MMLU-Pro (held-out) | **19.4** | 11.7 | 15.3 | 13.7 |
| Natural Questions (held-out) | 8.7 | 6.2 | 5.9 | **10.5** |
| TriviaQA (held-out) | **36.7** | 28.1 | 2.8 | 20.9 |
| GSM8K (5-shot) | 31.1 | 7.6 | 34.8 | **61.7** |
| MATH (4-shot) | 11.6 | 3.3 | 13.6 | **34.3** |
| HumanEval | 22.6 | 18.9 | 12.8 | **37.2** |

### A.7 Post-training

#### A.7.1 Dataset statistics

Table 10 shows the final composition of SmolTalk dataset.

Table 10: Composition of the SmolTalk dataset. The total dataset contains 1.1M instruction-response pairs from different data sources.

| Dataset source | Number of samples in SmolTalk |
| --- | --- |
| *New datasets* | |
| MagPie-Ultra | 431k |
| Smol-Rewrite | 56.2k |
| Smol-Constraints | 36.2k |
| Smol-Summarization | 101k |
| *Math datasets* | |
| NuminaMath-CoT | 112k |
| MetaMathQA | 50k |
| *Other* | |
| Self-OSS-Starcoder2-Instruct | 50.7k |
| APIGen-Function-Calling | 87.5k |
| SystemChats2.0 | 35.9k |
| LongAlign | 3.73k |
| Everyday-Conversations | 2.38k |
| Explore-Instruct-Rewriting | 32k |
| OpenHermes2.5 | 100k |
| **Total** | **1.1M** |

#### A.7.2 Evaluation of SmolTalk components

Table 13 shows the performance after training on the different components of SmolTalk we consider. The top section compares the results of fine-tuning SmolLM2 base on different instruction datasets, while the bottom section evaluates the impact of adding 20% specialized math data to a base mixture of 80% MagPie-Ultra+ during the SFT. The last row, SmolLM2-SFT, represents the final SFT checkpoint of SmolLM2 before DPO, trained for two epochs on the full SmolTalk dataset.

#### A.7.3 Instruct models evaluation

Table 12 compares the instruct variant of SmolLM2 to other instruction-tuned small models. Note that for Qwen2.5-1.5B-Instruct, we were unable to reproduce the authors' GSM8K and MATH scores, despite using their official math answer parser and chat template. As previously noted, we report all results using the same evaluation setup for consistency across models.

#### A.7.4 Specialized Reasoning Models

Recent work (Guo et al., 2025) explores building small models with strong reasoning abilities by specializing them in mathematics and code. For example, the DeepSeek-R1-Distill-Qwen-1.5B model (Guo et al., 2025) starts from Qwen2.5-Math-1.5B (Yang et al., 2024c), a continual pretraining of Qwen-2.5-1.5B on over a trillion tokens of curated math data, and is then fine-tuned on reasoning traces distilled from the larger DeepSeek-R1 model.

These approaches can achieve very high scores on math and code benchmarks but may come at the cost of general capabilities. As shown in Table 11, DeepSeek-R1-Distill-Qwen-

1.5B excels in mathematical reasoning but exhibits lower instruction-following capabilities (IFEval) compared to SmolLM2 and Qwen2.5-1.5B.

| Model | GSM8K | MATH | IFEval |
|---|---|---|---|
| SmolLM2-1.7B | 48.8 | 21.0 | **56.7** |
| Qwen2.5-1.5B | 63.3 | 19.6 | 47.4 |
| DeepSeek-R1-Distill-Qwen-1.5B | **84.3** | **86.3** | 45.1 |

Table 11: Performance comparison on GSM8K (mathematical reasoning) and IFEval (instruction following). Bold indicates best performance per benchmark.

Table 12: Comparison of 1-2B **instruction-tuned models** across benchmarks. SmolLM2-1.7B-Instruct exhibits strong performance in instruction-following, reasoning, and math.

| Model family | SmolLM2 | Llama3.2 | Falcon3 | Qwen2.5 |
|---|---|---|---|---|
| Parameters | 1.7B | 1.2B | 1.6B | 1.5B |
| IFEval (Average) | **56.7** | 53.5 | 56.5 | 47.4 |
| MT-Bench | 6.13 | 5.48 | 5.05 | **6.52** |
| OpenRewrite-Eval | 44.9 | 39.2 | 40.3 | **46.9** |
| ARC | **51.7** | 41.6 | 50.1 | 46.2 |
| MMLU-Pro | 19.3 | 12.7 | 15.8 | **24.2** |
| HellaSwag | **66.1** | 56.1 | 58.7 | 60.9 |
| PIQA | **74.4** | 72.3 | 72.1 | 73.2 |
| GSM8K (5-shot) | 48.8 | 37.4 | 41.1 | **63.3** |
| MATH (4-shot) | **21.0** | 19.5 | 17.5 | 19.6 |
| HumanEval | 28.1 | **33.5** | 11.6 | 30.5 |

Table 13: Performance on instruction-tuning datasets. MagPie-Ultra+ refers to MagPie-Ultra combined with Smol-Constraints, Smol-Rewrite, and Smol-Summarization. MagPie-Pro-MT is multi-turn while MagPie-Pro is the single turn version. All comparisons were performed by fine-tuning the SmolLM2 base model on each dataset for 1 epoch. SmolLM2-SFT[+], the final supervised fine-tuned version of SmolLM2, was trained for 2 epochs on SmolTalk.

| Dataset | IFEval | MTB | GSM8K | MATH | ARC-C | MMLU-Pro |
|---|---|---|---|---|---|---|
| *Instruction datasets comparison* | | | | | | |
| OpenHermes | 30.01 | 1.02 | **42.91** | 12.76 | 40.27 | **20.32** |
| UltraChat | 27.26 | 4.66 | 30.40 | 9.06 | **41.21** | 15.79 |
| MagPie-Pro | 30.45 | 4.31 | 14.56 | 6.64 | 36.01 | 12.19 |
| MagPie-Pro-MT | 31.66 | **5.40** | 20.55 | 7.84 | 36.69 | 11.97 |
| MagPie-Ultra | 35.49 | 5.22 | 24.34 | **13.56** | 37.71 | 12.01 |
| MagPie-Ultra+ | **48.16** | 5.28 | 19.94 | 12.74 | 38.91 | 12.43 |
| *Math datasets comparison* | | | | | | |
| MagPie-Ultra+ + MathInstruct | **47.05** | 5.43 | 30.1 | 14.0 | **38.99** | **13.65** |
| MagPie-Ultra+ + MetaMathQA | 44.98 | 5.02 | **47.08** | 17.56 | 36.77 | 12.18 |
| MagPie-Ultra+ + NuminaMath-CoT | 46.27 | **5.99** | 25.32 | **18.00** | 37.88 | 12.58 |
| *Full SmolTalk* | | | | | | |
| SmolTalk | 46.67 | 5.49 | 43.75 | 18.60 | 40.02 | 18.19 |
| SmolLM2-SFT[+] | **57.09** | **6.11** | **47.54** | **19.64** | **42.49** | **19.06** |

### A.8 Long context

#### A.8.1 Training

To support long-context applications, we followed standard practice (Gao et al., 2024) and extended the context length from 2k to 8k tokens, by taking an intermediate checkpoint from stage 4 (before the final 75 billion tokens of training) and continuing training with a different data mixture and a RoPE value of 130k. The mixture was adjusted to include 40% long-context documents (8k tokens or more) sourced from DCLM (10%), FineWeb-Edu (10%), and the books subset of Dolma (20%) (Soldaini et al., 2024), while the remaining 60% followed the stage 4 mixture.

#### A.8.2 Evaluation

Figure 8 shows the evaluation results on the Needle in the Haystack benchmark and Table 14 shows the evaluation results on the HELMET benchmark.

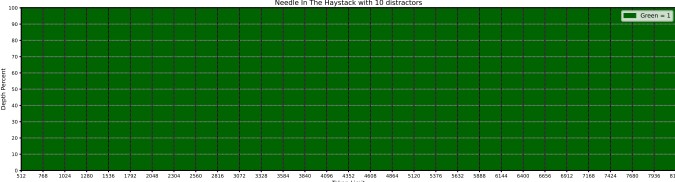

Figure 8: Needle in the Haystack evaluation of SmolLM2 with 8192 context length.

Table 14: Evaluation results of the base models on the HELMET benchmark using 8k maximum input length.

| Metric | SmolLM2-1.7B | Llama3.2-1B | Qwen2.5-1.5B |
|---|---|---|---|
| Recall | 36.38 | 55.81 | **66.94** |
| RAG | 47.17 | 42.13 | **47.54** |
| ICL | 23.20 | 51.20 | **52.00** |
| Re-rank | 23.31 | 26.93 | **29.29** |
| LongQA | **33.00** | 21.99 | 26.23 |

