# OpenReview forum: "SmolLM2: When Smol Goes Big — Data-Centric Training of a Fully Open Small Language Model"
_colmweb.org/COLM/2025/Conference — COLM 2025_

### Official Review · Reviewer_sj9u · 2025-04-25

**Rating:** 7
**Confidence:** 5
**Ethics Flag:** 1

**Summary:**

This work builds a fully transparent (code, parameters, and training data) small (1.7B) language model, SmolLM2-1.7B, which undergoes both pretraining and posttraining. To build the model, this work also creates (openly available) high-quality training datasets for mathematical reasoning, code reasoning, and instruction following. Experimental results show that SmolLM2-1.7B outperforms LLaMA 3.2-1B, Falcon 3-1.6B, and Qwen 2.5-1B on several evaluations.

**Questions To Authors:**

Could you provide more details about your evaluation using MT-Bench? For example, which LLM did you use as the judge?

**Reasons To Accept:**

+ Building open models is always appreciated, especially opening the training data, which is increasingly recognized as crucial for future development.
+ The release of high-quality training datasets is very valuable for the community.
+ The performance of SmolLM2-1.7B seems to be promising and competitive.
+ This paper provides detailed analysis and practical guidance on how to construct effective data mixtures, which is quite insightful.

**Reasons To Reject:**

In general, the evaluations could be more comprehensive.
+ To evaluate and compare a group of instruction-tuned models, it would be better to compute the win rate (as determined by both human raters and LLM-as-a-judge) of each model against a fixed baseline (e.g., LLaMA 3.1 405B Instruct).
+ It would also be interesting to compare SmolLM2-1.7B with DeepSeek-R1-Distill-Qwen-1.5B. Although it is expected that the latter would outperform on reasoning tasks, this is acceptable since that model is not fully open and uses significantly different techniques.

Additionally, it would be helpful to include a more detailed analysis of the training compute. Providing more information about the infrastructure would also benefit readers.

---

> ### Author Response · Authors · 2025-06-02
>
> We would like to thank reviewer `sj9u` for their detailed review and feedback regarding the evaluation setup. We address the points raised by the reviewer below.
>
> ---
>
> ## Evaluation
>
> ### Win Rate
> > To evaluate and compare a group of instruction-tuned models, it would be better to compute the win rate (as determined by both human raters and LLM-as-a-judge) of each model against a fixed baseline (e.g., Llama 3.1 405B Instruct).
>
> We were also curious about the results of human evaluation, so we submitted the model to the LMSYS Chatbot Arena leaderboard (https://arxiv.org/pdf/2403.04132), where models are compared through pairwise evaluations judged by human annotators. SmolLM2 obtained an Elo score of `1043` (95% CI: +15/-12); for comparison, Llama3.2-1B-Instruct scores `1050` (95% CI: +6/-6). We would be glad to include this result in a future revision.
>
> ### MT-Bench Setup
>
> > Could you provide more details about your evaluation using MT-Bench? For example, which LLM did you use as the judge?
>
> We followed the official MT-Bench evaluation setup from LMSYS (Zheng et al., 2023) for the questions and judge prompts. We used `GPT-4-0613` as the judge and `GPT-3.5-Turbo-0125` to generate the reference answers. We will include these clarifications in the paper version.
>
> ### DeepSeek Comparison
>
> > It would also be interesting to compare SmolLM2-1.7B with DeepSeek-R1-Distill-Qwen-1.5B. Although it is expected that the latter would outperform on reasoning tasks, this is acceptable since that model is not fully open and uses significantly different techniques.
>
> As the reviewer mentions, the training of DeepSeek-R1-Distill-Qwen-1.5B differs substantially in setup compared to other instruction-tuned models. Beyond distillation on reasoning traces, it is finetuned from Qwen2.5-Math-1.5B, a continual pretraining of Qwen2.5-1.5B on trillions of math tokens. As such, while it would likely outperform SmolLM2 on math and reasoning benchmarks, we expect different trade-offs on broader tasks.
>
> ---
>
> ## Compute Details
>
> > Additionally, it would be helpful to include a more detailed analysis of the training compute. Providing more information about the infrastructure would also benefit readers.
>
> We will include these estimates in a supplemental table to improve transparency and reproducibility. SmolLM2 was pretrained over 24 days on 256 `H100` GPUs (660W TDP), which amounts to roughly `147,000 GPU-hours`, excluding ablation experiments. Additionally, we conducted around 25 smaller-scale ablation runs, estimated to account for another `65,000 GPU-hours`.

---

> > ### Comment · Reviewer_sj9u · 2025-06-02
> >
> > Thanks for your response!
> >
> > For DeepSeek comparison, I agree that "while it would likely outperform SmolLM2 on math and reasoning benchmarks, we expect different trade-offs on broader tasks", so that's why I was asking for the empirical results.

---

> > > ### Author Response · Authors · 2025-06-10
> > >
> > > Thank you for your reply! We'll include a performance comparison with DeepSeek-Distill-Qwen-1.5B in the appendix of a revision, to illustrate the trade-offs in performance.

---

### Official Review · Reviewer_8zt8 · 2025-05-12

**Rating:** 9
**Confidence:** 4
**Ethics Flag:** 1

**Summary:**

This paper documents and publicly releases SmolLM2, a fully transparent state-of-the-art ``small'' (1.7 billion parameter) language model (LM), along with its training datasets and code. SmolLM2 is trained on 11 trillion tokens of data using mutl-stage training process. The paper also conducted extensive ablation studies on SmolLM2 to provides the insights on different components of the model.

**Reasons To Accept:**

1. The model and datasts released is of great importance to the LLM community and shed light on the future research and applications of small LLMs.
2. The experiments are insightful and comprehensive.
3. The paper is well written and easy to follow.

**Reasons To Reject:**

1. There are barely any new techniques proposed.

---

> ### Author Response · Authors · 2025-06-02
>
> We thank the reviewer for the encouraging feedback. We're glad that the contributions, experiments, and released resources were appreciated, and we hope SmolLM2 and its datasets prove valuable to the community.

---

> > ### Comment · Reviewer_8zt8 · 2025-06-10
> >
> > Thank you for your response. I would like to maintain my score.

---

### Official Review · Reviewer_FMrj · 2025-05-12

**Rating:** 7
**Confidence:** 4
**Ethics Flag:** 1

**Summary:**

This paper presents SomlLM2, a fully transparent small language model. SmolLM2 is trained on ~11 trillion tokens of data with multi-stage training process. Different data mixtures are applied during different training stages. The authors demonstrate SmolLM2 outperforms other small languages at similar scale such as Qwen-2.5-1B, Llama-3.2-1B, etc.
The paper itself, although does not propose any new model architecture, nor training algorithms, contains several new training datasets such as FineMath, Stack-Edu and SmolTalk, as well as how these datasets are mixed during the pretraining and post-training process. These information, I believe, is beneficial and interesting to the COLM community.

**Questions To Authors:**

- I particularly like the "online" approach that dynamically adjust data mixture across different stages. If I understood it correctly, this also brings more hyper-parameters on the data side, e.g., how many tokens to allocate to each stage. I understand grid searching these parameters is prohibitively expensive, so I wonder how the authors select these parameters: why 6T to 8T tokens in stage 2 for example? Where 4 stages? Is it somewhat heuristic, or there exists a more principled approach?
- I wonder if the authors could comment on the new challenges brought by this online dynamic data mixture. Assuming that a more fine-grained or even continuous stages could potentially improve the performance of SLMs/LLMs a lot, what are the new challenges that we need to overcome?

**Reasons To Accept:**

1. The topic is timely and highly relevant to the COLM community.
2. The authors presents three new datasets that outperforms existing datasets on various downstream tasks: FineMath, Stack-Edu and SmolTalk. I think these datasets will be widely interesting to the ML/LLM community.
3. The authors shows an "online" training paradigm where the data mixture is adjusted at different stages. The authors presents detailed, transparent data mixture strategies across different stages, which is particularly interesting to the community.
4. The dataset, trained models, code are made openly available, which could facilitate future research and application of small language models.

**Reasons To Reject:**

- I found the selection of data in the post-training step rather random or unjustified. For example, what is the rationale behind using `Self-OSS-Starcoder2-Instruct`? If the authors have tried other open datasets, could the authors disclose what options they tried and why this particular dataset stands out? I believe such information will bring more insights to the research community.
- The authors mentioned: "First, we performed a careful evaluation of existing web, code, math, and instruction-following datasets (Section 3) to guide training data design choices. After finding that existing datasets were too small and/or low-quality, we created new state-of-the-art datasets: FineMath, Stack-Edu, and SmolTalk (for mathematics, code, and instruction-following respectively)". As I understood, one main contribution of this paper is these three transparent datasets. However,I felt the presentation of the three new datasets can be improved. I would personally prefer a table with concrete characteristics of each datasets. For example,

| name     | #Samples | #Tokens | Performance Summary |   |
|----------|----------|---------|---------------------|---|
| FineMath |     6.7M     |   10B      |  ...                   |   |

It's not a show-stopper though.

- For the English Web Data, I felt some important baselines are missing. The authors mentioned two notable open datasets: FineWeb-Edu and DCLM, without discussing or mentioning other open datasets such as RedPajama, LLM 360, etc. I wonder what is the main rationale behind this and why the authors do not compare against them.

---

> ### Author Response · Authors · 2025-06-02
>
> We would like to thank the reviewer for the thoughtful feedback, and for mentioning relevant points that can be made clearer in the paper. We address the concerns raised by the reviewer below.
>
> ---
>
> ## Post-training ablations
>
> > I found the selection of data in the post-training step rather random or unjustified. For example, what is the rationale behind using Self-OSS-Starcoder2-Instruct? If the authors have tried other open datasets, could the authors disclose what options they tried and why this particular dataset stands out? I believe such information will bring more insights to the research community.
>
> We agree that clarification is needed and thank the reviewer for pointing this out. While we reported ablations for the Math and English SFT datasets (Section 5.1, Table 11), we did not include similar details for the other specialized subsets. For code, we compared Self-OSS-Starcoder2-Instruct, Code-Feedback ([Zheng et al., 2024](https://arxiv.org/abs/2402.14658)), and MagiCoder ([Wei et al., 2023](https://arxiv.org/abs/2312.02120)), selecting the first based on HumanEval performance. For long context, we evaluated LongAlign and SeaLong ([Li et al., 2024](https://arxiv.org/abs/2411.08147v1)), with LongAlign providing better scores on HELMET. For Smol-Constraint, Smol-Rewrite, and Smol-Summarization, no suitable open SFT datasets existed, so we created and validated them (Table 11). We will include this in a future revision.
>
> ---
>
> ## Datasets
>
> > As I understood, one main contribution of this paper is these three transparent datasets. However,I felt the presentation of the three new datasets can be improved. I would personally prefer a table with concrete characteristics of each datasets.
>
> A table would indeed better showcase the characteristics of each dataset, and we will include this in a revision.
>
> > For the English Web Data, I felt some important baselines are missing. The authors mentioned two notable open datasets: FineWeb-Edu and DCLM, without discussing or mentioning other open datasets such as RedPajama, LLM 360, etc. I wonder what is the main rationale behind this and why the authors do not compare against them.
>
> We focused on DCLM and FineWeb-Edu as they currently represent the strongest open web datasets for LLM pretraining, as shown in their respective papers. Both substantially outperform prior baselines such as RedPajama, LLM360, and Dolma. For instance, FineWeb-Edu outperforms the next best dataset (Matrix) by 3.9% on average over 8 tasks ([Penedo et al., 2024](https://arxiv.org/abs/2406.17557)) for a 1.7B model, and the DCLM paper ([Li et al., 2024](https://arxiv.org/abs/2406.11794)) reports a 16.7% improvement over FineWeb-Edu on average across 22 tasks for a 7B model. However, in our experiments, we did not observe DCLM consistently outperforming FineWeb-Edu, so we included both datasets.
>
> ---
>
> ### Multi-stage training
>
> > I particularly like the "online" approach that dynamically adjusts data mixture across different stages. If I understood it correctly, this also brings more hyper-parameters on the data side, e.g., how many tokens to allocate to each stage. I understand grid searching these parameters is prohibitively expensive, so I wonder how the authors select these parameters: why 6T to 8T tokens in stage 2 for example? Where 4 stages? Is it somewhat heuristic, or there exists a more principled approach?
>
> We selected these hyperparameters based on the principles outlined in Section 4, in particular performance-driven interventions and minimizing data repetition. By 8T tokens, StarCoderData had undergone ~4 epochs (1T tokens) and OWM showed limited math improvements after ~1 epoch on the dataset as shown in table 4, prompting a switch to new and higher-quality datasets like Stack-Edu and InfiMM-WebMath. Stage 3 was then conducted until 10T tokens, allowing Stage 4 to span from 10T to 11T tokens. This ensures the final decay phase covers 10% of training, following WSD schedule recommendations ([Hägele et al., 2024](https://arxiv.org/pdf/2405.18392)). While this improved performance, we acknowledge that other configurations might also be effective and do not claim our setup to be globally optimal.
>
> > I wonder if the authors could comment on the new challenges brought by this online dynamic data mixture. Assuming that a more fine-grained or even continuous stages could potentially improve the performance of SLMs/LLMs a lot, what are the new challenges that we need to overcome?
>
> Frequent data mixture changes risk overfitting to short-term metric fluctuations or noise. We believe that having two stages—a stable phase and a decay phase with high-quality data—followed by few targeted adjustments during the stable phase when performance plateaus strikes a good balance without increasing training complexity. To mitigate overfitting, it is critical to use held-out benchmarks that are not involved in training decisions. We will briefly emphasize this perspective in an updated revision.

---

> ### Comment · Reviewer_FMrj · 2025-06-09
> **Thanks for the clarification**
>
> Thank you for the clarification. I will raise my score up :).

---

### Official Review · Reviewer_fgjE · 2025-05-13

**Rating:** 6
**Confidence:** 3
**Ethics Flag:** 1

**Summary:**

This paper introduces SmolLM2, a 1.7B-parameter language model designed to match or outperform other small LMs like Qwen2.5-1.5B and Llama3.2-1B, with full transparency of training data, methodology, and code. The authors tackle the lack of reproducibility in existing small LMs by releasing not only the model but also the associated datasets and training process.

Key contributions include:

* Full release of training datasets (\~11T tokens), pipeline, and codebase.
* Introduction of three new datasets: FineMath, Stack-Edu, and SmolTalk, aimed at improving math, code, and instruction-following capabilities.
* A manual rebalancing strategy across training stages using performance-based mixing rate adjustments.
* Extensive ablations and benchmark evaluations (e.g., MMLU, GSM8K, HumanEval) demonstrate that SmolLM2 is state-of-the-art for its size.
* Encourages reproducibility and downstream research by making all assets public.




The paper makes a clear, original, and impactful contribution to the area of small language models by addressing the major limitation of transparency.
* The empirical results are thorough and convincingly demonstrate that SmolLM2 is state-of-the-art among similar-sized models.
* The new datasets and documented pipeline will likely become important benchmarks and tools for the community.
* While not introducing a novel model architecture, the work’s focus on data, evaluation, and reproducibility fills a critical and underserved niche.

**Reasons To Accept:**

1. The release of training data and code is a major differentiator in the current opaque landscape of small LMs.
2. The paper includes rigorous ablations and benchmark results comparing datasets and training regimes.
3. FineMath significantly improves math reasoning over existing datasets like OpenWebMath. Stack-Edu and SmolTalk fill gaps in code and instruction-following datasets.
4. The focus on small models is timely and practical for edge-device deployment.
5. Ablation studies and data-driven design choices enhance the credibility of the reported performance gains.

**Reasons To Reject:**

1. While well-motivated, the manual tuning of mixture weights is not easily reproducible or scalable. An automated, learnable mixing approach could improve robustness and generalizability.
2. The model architecture follows standard LLaMA-based Transformer setups. The novelty lies primarily in the data and training procedure, which may be seen as incremental by some.
3. Despite the open release, training on 11T tokens (\$250K compute) sets a high entry bar for true replication.

---

> ### Author Response · Authors · 2025-06-02
>
> We thank reviewer `fgjE` for their thoughtful review and constructive feedback. We appreciate the recognition of our contributions, especially regarding transparency and dataset releases. Below, we address the reviewer’s concerns.
>
> ---
>
> > While well-motivated, the manual tuning of mixture weights is not easily reproducible or scalable
>
> While our tuning of mixture weights was largely done heuristically based on intermediate model performance, we note that automated data mixture optimization methods have not proven effective at large-scale pretraining, and manual annealing experiments have been successfully employed in other large-scale model trainings such as Llama3 (https://arxiv.org/abs/2407.21783) and Olmo2 (https://arxiv.org/abs/2501.00656). To ensure reproducibility, we will release training scripts and full details.
>
> ---
>
> > *The model architecture follows standard LLaMA-based Transformer setups. The novelty lies primarily in the data and training procedure, which may be seen as incremental by some.*
>
> We acknowledge the limited architectural novelty. However, this choice is intentional—by using a standard architecture, we demonstrate that performance gains stem primarily from our data curation and multi-stage training approach. Our results show this strategy can match/exceed SOTA models (Qwen2.5-1.5B, Llama3.2-1B, Falcon3-1.6B) with closed training details, providing a path to competitive performance.
>
> ---
>
> > *Despite the open release, training on 11T tokens ($250K compute) sets a high entry bar for true replication.*
>
> The reviewer raises an important concern about the high computational cost barrier for replication. To improve accessibility, we trained smaller 135M and 360M parameter models that achieve state-of-the-art performance in their size, for less than $18K for the 135M model. These models will be included in the additional page available in the paper revision. We believe that resource-intensive research remains highly valuable, especially when it is accompanied by full transparency—something that is unfortunately rarely shared in the literature, as the reviewer points out.

---

### Decision · Program_Chairs · 2025-07-08

**Decision:**

Accept

**Comment:**

This paper introduces a new 1.7B model, high-quality pre-training data at an impressive scale (11T tokens) and detailed ablations and benchmark evaluations showing a very strong new model and dataset compared to the previous state of the art (DCLM). The release of pre-training data and full codebase is very valuable for open-sourcing LM research.

Some concerns were raised about the difficulty of replicating due to high cost but this is unavoidable due to the nature of the research and should not be held against the authors.
Some additional concerns were raised about the selection of some post-training datasets and hyper-parameters of the data mixture, but again, in my opinion the contribution is very valuable and the concerns are minimal and well addressed by the authors.

Overall a great contribution with detailed scientific exploration of LM design choices and valuable code and dataset releases.